# Genetic Testing for Patients with Cardiomyopathies: The INDACO Study—Towards a Cardiogenetic Clinic

Matteo Bianco [1,*,†], Noemi Giordano [2,†], Valentina Gazzola [1], Carloalberto Biolè [1], Giulia Nangeroni [1], Maurizio Lazzero [1], Giulia Margherita Brach del Prever [3], Fiorenza Mioli [3], Giulia Gobello [1], Amir Hassan Mousavi [1], Monica Guidante [2], Silvia Deaglio [3], Daniela Francesca Giachino [2,‡] and Alessandra Chinaglia [1,‡]

1   Cardiology Division, San Luigi Gonzaga University Hospital, Regione Gonzole 10, 10043 Turin, Italy; valentina.gazzola@edu.unito.it (V.G.); carloalberto.biole@gmail.com (C.B.); giulia.nange@gmail.com (G.N.); lazzerom@gmail.com (M.L.); gobello.giulia@gmail.com (G.G.); amir.mousavi@edu.unito.it (A.H.M.); a.chinaglia@sanluigi.piemonte.it (A.C.)
2   Medical Genetics, Department of Clinical and Biological Sciences, San Luigi Gonzaga University Hospital, University of Turin, 10043 Orbassano, Italy; n.giordano@unito.it (N.G.); monica.guidante@edu.unito.it (M.G.); daniela.giachino@unito.it (D.F.G.)
3   Immunogenetics and Biology Transplant Service, Department of Medical Sciences, University Hospital "Città della Salute e della Scienza di Torino", University of Torino, 10100 Turin, Italy; giuliamargherita.brachdelprever@unito.it (G.M.B.d.P.); fiorenza.mioli@unito.it (F.M.); silvia.deaglio@unito.it (S.D.)
*   Correspondence: matteo.bianco87@gmail.com; Tel.: +39-011-9026706
†   Joined first authorship.
‡   Joined last authorship.

**Abstract:** Cardiomyopathies have evolved from being considered rare and idiopathic to being increasingly linked to genetic factors. This shift was enabled by advancements in understanding genetic variants and the widespread use of next generation sequencing (NGS). Current guidelines emphasize the importance of evidence-based gene panels that can offer "clinically actionable results", which provide diagnostic and prognostic insights. They also advise against indiscriminate family screening after finding variants of uncertain significance (VUS) and recommend collaboration among multidisciplinary teams for an accurate variant pathogenicity assessment. This article presents an innovative "cardiogenetic clinic" approach involving cardiologists and medical geneticists to provide genetic testing and family screening. This study attempts to improve the diagnostic process for suspected genetic cardiomyopathies; this includes direct patient recruitment during cardiology appointments, NGS analysis, and combined consultations with cardiologists and geneticists to assess the results and screen the families. The study cohort of 170 patients underwent genetic testing, which identified 78 gene variants. Positive results (C4 or C5 variants) occurred in 20 (19.8%) cases, with rates varying by cardiomyopathy phenotype, while 57 (73.1%) of the variants found were classified as C3-VUS, causing a significant management issue. This model shortened the time to results, increased patient adherence, and improved patients' diagnoses. Family screening was pondered depending on the relevance of the detected variants, showing this method's potential to impact patient management.

**Keywords:** cardiomyopathies; genetic testing; next generation sequencing (NGS)

## 1. Introduction

Cardiomyopathies are one of the leading causes of heart failure and sudden cardiac death in the world. In the past, cardiomyopathies were overall considered rare diseases, primarily diagnosed by exclusion of other causes of heart failure and mostly classified as "idiopathic" [1]. Nowadays, even though the exclusion of other causes of heart failure remains of paramount importance in the diagnostic workup of cardiomyopathies, they are being increasingly linked to their genetic background. This has become possible thanks to

the advancements in knowledge of disease-causing genetic variants and to the widespread diffusion of new genetic testing strategies.

Primary genetic cardiomyopathies are emerging from the group that used to be defined as "idiopathic" thanks to the identification of the individual genes most commonly responsible for each phenotype. Even though significant overlap exists, i.e., the same gene can potentially be involved in more than one phenotype, primary cardiomyopathies are monogenic diseases (caused by variations in single genes). To date, variants in over 100 genes are known to be related to cardiomyopathies [2]. This process of defining disease-causing gene variants—albeit far from being concluded—has become possible largely owing to the diffusion of next generation sequencing (NGS). NGS is a sequencing technique that allows a parallel examination of large panels of genes, making DNA sequencing significantly quicker and more cost-effective.

The current position paper regarding appropriate gene panels, presented by the European Society of Cardiology in 2022, reinforces the basic principle that all tested genes should have strong evidence supporting their causal link with the disease in order to avoid as much as possible giving patients—and their families—wrong or uncertain information (which may, in certain cases, even prompt unnecessary treatment, especially for family members). They also state that a "clinically actionable result" (i.e., a pathogenic or likely pathogenic variant—meaning a variant with >90% certainty of being disease-causing) can not only provide diagnostic confirmation in the proband but also give prognostic and therapeutic information for many phenotypes and elicit the genetic screening of family members at risk. On the contrary, they do not recommend indiscriminate family screening after finding variants of uncertain significance (VUSs); rather, they suggest that clinical screening should be employed and that disease-specific multi-disciplinary teams should get together to help classify the real pathogenicity of the variant. Moreover, it is recommended that patients and their families are informed of the inheritance mode of the disease and that being carrier of a variant does not imply showing clinical signs and symptoms of the disease with absolute certainty. All this information can also be relevant in the setting of reproductive counselling for couples who are carriers of a variant [3].

In light of the relevance that the genetic background of cardiomyopathies is assuming, it is logical that great awareness is rising towards optimizing patients' access to genetic testing, improving the interpretation of results and the management of at-risk family members. In order to face these rising challenges, there is an increasing need for specialized programs that integrate clinical cardiovascular medicine and genetic expertise [4].

The aim of the present study, the "Iter diagnostico dei pazienti con sospetto di cardiomiopatie geneticamente determinate: ottimizzare tempi e risorse"—INDACO study, a monocenter prospective study, is to evaluate the workup of patients with cardiomyopathies suspected of having a genetic etiology in the regional context of Piedmont (northern Italy). Specifically, the target is the implementation of a more functional workup process, pursuing time and resource optimization through the creation of a specific pathway for patients with cardiomyopathies inside a "cardiogenetic clinic" in which cardiologists and medical geneticists work together in a more functional way in order to provide timely and effective genetic testing and results, other than prompt screening of family members when indicated.

## 2. Materials and Methods

The INDACO study prospectively included all the patients from the Cardiology Division of San Luigi Gonzaga University Hospital with a suspected genetic cardiomyopathy who underwent genetic testing with NGS from December 2021 to December 2023 and who were managed with our new workup. The study was performed according to the San Luigi Gonzaga University Hospital Review Board guidance and was conducted in accordance with the Declaration of Helsinki and its later amendments. All patients signed an informed consent form and all the data were anonymized before being collected.

In the past, the procedure for patients to access genetic testing in our hospital started with a first referral to the medical geneticist by the cardiologist once the suspicion of

a genetically determined cardiomyopathy arose. The genetic test could be performed only after an initial consultation with the medical geneticist, and an additional genetic consultation was also required to interpret the results before going back to the cardiologist for the management of clinical implications (if any). Alternatively, a minority of patients were referred for genetic testing directly after having reached the attention of medical geneticists in the presence of complex or syndromic clinical pictures possibly involving the heart and were only referred for a cardiological evaluation afterwards (when relevant). Overall, this procedure was lacking standardization and, as a result, it led to a considerable number of patients not complying with the multiple referrals for consultation with different specialists. Additionally, many other cardiology departments face challenges in managing genetic consultations performed by cardiologists.

With our new workup, the patients are referred to our cardiogenetic outpatient's clinic from the cardiomyopathy clinic or following hospitalization in the cardiology department due to acute cardiac events. All the patients are > 18 years old.

During the first consultation prior to testing, the clinical cardiologist comprehensively evaluates the patient's history and clinical picture to exclude other etiologies possibly accounting for the patient's phenotype. Moreover, diagnostic procedures have been performed to exclude phenocopies. For instance, in order to rule out the possibility of Brugada syndrome, an MRI and, if necessary, an ajmaline test are included in the full workup for arrhythmogenic cardiomyopathy. Patients are informed about the type of DNA test they will undergo, including the genes studied and the technologies used. They are also made aware of the potential clinical benefits and turnaround time [5] so that they can make informed decisions [6,7]. Patients are usually considered for NGS when they have a significant family history of cardiomyopathies and/or when a clear explanation for their clinical picture is lacking. In the context of this first consultation, patients who give their informed consent undergo a blood draw and the collected sample is then sent directly to the reference center (S.C.U. of Immunogenetic and Transplant Biology of AOU Città della Salute e della Scienza, Turin) for the NGS analysis. The request for the genetic test is made by the cardiologist through the online platform of the Piedmont Regional Transplant Center (www.cse.crtpiemonte.it, accessed on 30 April 2024), which is already known and well established for the diagnostic flow of monogenic kidney diseases [8], in a specific section dedicated to genetic counseling for NGS of hereditary diseases. In this section, the cardiologist provides patient's generalities and relevant information about his or her clinical picture and family history to the medical genetic center performing the genetic analysis. Furthermore, a full clinical report (normally the first visit made by a cardiologist in the cardiogenetic clinic) must be uploaded, together with an electronic prescription for the genetic evaluation. After the request has been uploaded, it is evaluated by a medical geneticist, and a confirmation of the acceptance is sent back to the cardiologist. A preliminary consultation with the geneticist is instead reserved for patients whose diagnostic hypothesis and phenotype are particularly complex to verify the appropriateness of the genetic testing request and also to optimize NGS gene targets. The sequences are aligned to the human reference genome GRCh37. Variants are annotated according to the HGVS nomenclature and classified according to the 2015 American College of Medical Genetics and Genomics (ACMG) guidelines. Variants of uncertain significance associated with pathologies with an autosomal recessive mode of inheritance, variants classified as benign or probably benign, and synonymous variants that do not impact splicing are not reported; hence, the presence of C1 or C2 variants is considered as a negative result. When C3 variants (VUSs, variants of unknown significance) are identified, the test is considered inconclusive. Once the analysis is complete, the results are uploaded on the same online platform and the cardiologists are notified. All patients with a positive test result (i.e., if C5 or C4 variants are found) or with a C3-VUS result are offered a joint consultation with the cardiologist and the medical geneticist to discuss the report. During this consultation, further details about the patient's family history are collected, the pedigree is drawn by the medical geneticist, and the possibility of testing other family members (with Sanger

sequencing) is evaluated. Patients with negative test results are instead directly informed by the cardiologists during routine follow-up visits.

After finding a VUS, the family history of the patient is carefully re-evaluated and, in case of a solid clinical suspicion emerging from a familial evaluation, first-degree relatives are screened from the clinical point of view in order to investigate the presence of any unrecognized cardiomyopathy phenotype. Genetic testing (i.e., Sanger sequencing for relevant genes) is then only performed on individuals showing the phenotype, with the aim of familial segregation. The patient's workup is summarized in Figure 1.

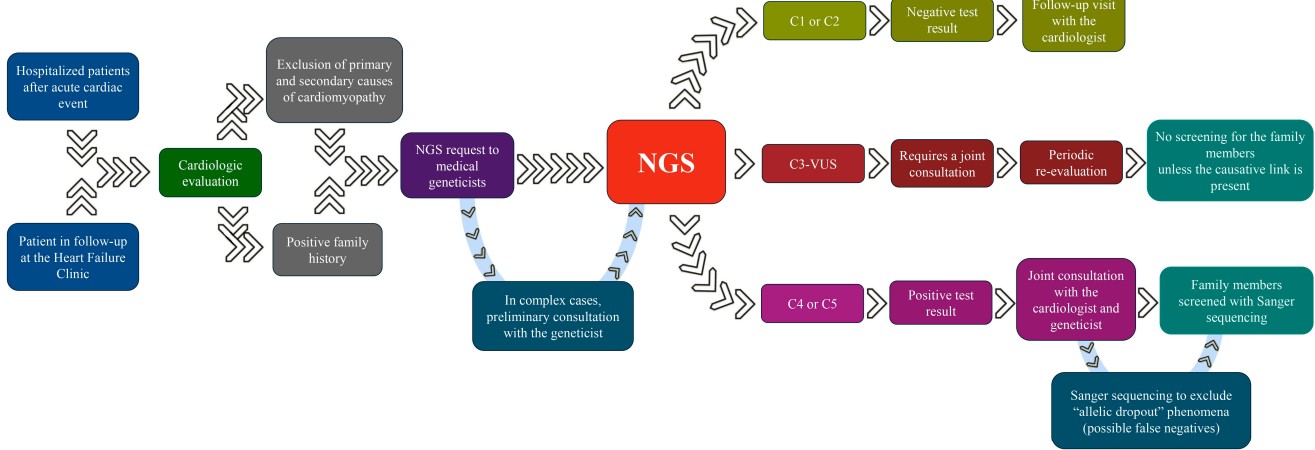

**Figure 1.** INDACO's patient workup.

The NGS panels employed by the laboratory were either TruSight Cardio (Illumina) (Table 1) or TruSight One Expanded (Table 2), a more extended panel, or OMIM, PanelApp and Orphanet databases were used to identify the genes associated with the clinical indication, and hence, the tested gene panels have been evolving over time parallel to the latest findings in the field. Moreover, additional genes can be added to the panels based on specific clinical indications for each individual patient. It is also possible to test multiple panels for the same patient, in accordance with precise indications by the medical geneticists or, in selected cases, the whole clinical exome is sequenced.

**Table 1.** Next generation sequencing (NGS) using True Sight Cardio panels. Abbreviations: DCM, dilated cardiomyopathy; HCM, hypertrophic cardiomyopathy; ARVD, arrhythmogenic right ventricular cardiomyopathy; LVNC, left ventricular non-compaction cardiomyopathy.

| Panel of 64 Genes DCM and ARVD | Panel of 50 Genes for HCM | Panel of 17 Genes for LVNC |
|---|---|---|
| *ABCC9; ACTA1; ACTC1; ACTN2; ALMS1; ANK2; ANKRD1; BAG3; CRYAB; CSRP3; DES; DMD; DNAJC19; DOLK; DSC2; DSG2; DSP; EMD; EYA4; FHL1; FHL2; FKRP; FKTN; GATAD1; GLA; HFE; ILK; JUP; LAMA4; LAMP2; LDB3; LMNA; MYBPC3; MYH6; MYH7; MYL2; MYL3; MYPN; NEXN; NKX2-5; PDLIM3; PKP2; PLN; PRDM16; RAF1; RBM20; RYR2; SCN1B; SCN5A; SDHA; SGCD; TAZ; TBX5; TCAP; TGFB3; TMEM43; TNNC1; TNNI3; TNNT2; TPM1; TTN; TTR; TXNRD2; VCL.* | *ABCC9; ACTC1; ACTN2; BAG3; BRAF; CACNA1C; CALR3; CAV3; COX15; CRYAB; CSRP3; DES; FHL1; FXN; GAA; GLA; HRAS; JPH2; KLF10; LAMP2; LDB3; MAP2K1; MAP2K2; MYBPC3; MYH6; MYH7; MYL2; MYL3; MYLK2; MYO6; MYOZ2; MYPN; NEXN; NRAS; PLN; PRKAG2; PTPN11; RAF1; SHOC2; SLC25A4; SOS1; TCAP; TNNC1; TNNI3; TNNT2; TPM1; TRIM63; TTN; TTR; VCL* | *ACTC1; CASQ2; DNAJC19; DTNA; HCN4; LDB3; MIB1; MYBPC3; MYH7; NKX2-5; PRDM16; RYR2; TAZ; TBX5; TNNT2; TPM1; TTN.* |

**Table 2.** Next generation sequencing (NGS) using True Sight One expanded panels. Abbreviations: DCM, dilated cardiomyopathy; HCM, hypertrophic cardiomyopathy; ARVD, arrhythmogenic right ventricular cardiomyopathy; LVNC, left ventricular non-compaction cardiomyopathy.

| Panel of 12 Genes for Amyloidosis | Panel of 58 Genes for HCM | Panel of 79 Genes for DCM and ARVD |
|---|---|---|
| APOA1, APOA2, APOC2, APOC3, B2M, FGA, GLA, GSN, LYZ, MEFV, NLRP3, TTR | ABCC9, ACADVL, ACTC1, ACTN2, AGL, ATAD3A, BAG3, BRAF, CACNA1C, CALR3, CAV3, COX15, CRYAB, CSRP3, DES, FHL1, FHOD3, FLNC, FXN, GAA, GLA, GYG1, HRAS, JPH2, KLF10, LAMP2, LDB3, LZTR1, MAP2K1, MAP2K2, MYBPC3, MYH6, MYH7, MYL2, MYL3, MYLK2, MYO6, MYOZ2, MYPN, NEXN, NRAS, PLN, PRKAG2, PTPN11, RAF1, RIT1, SHOC2, SLC25A4, SOS1, TCAP, TNNC1, TNNI3, TNNT2, TPM1, TRIM63, TTN, TTR, VCL | ABCC9, ACTA1, ACTC1, ACTN2, ALMS1, ANK2, ANKRD1, BAG3, CDH2, CRYAB, CSRP3, CTNNA3, DES, DMD, DNAJC19, DOLK, DSC2, DSG2, DSP, EMD, EPG5, EYA4, FHL1, FHL2, FKRP, FKTN, FLNC, GATAD1, GLA, HAMP, HFE, HFE2, IDH2, ILK, JUP, LAMA4, LAMP2, LDB3, LMNA, MYBPC3, MYH6, MYH7, MYL2, MYL3, MYPN, NEXN, NKX2-5, PDLIM3, PKP2, PLN, PRDM16, PSEN1, PSEN2, RAF1, RBM20, RYR2, SCN1B, SCN5A, SDHA, SGCD, SLC40A1, SLC6A6, SPEG, TAZ, TBX5, TCAP, TFR2, TGFB3, TMEM43, TNNC1, TNNI3, TNNI3K, TNNT2, TPM1, TTN, TTR, TXNRD2, VCL, XK |

Furthermore, 10 patients who underwent NGS were afterwards also subject to Sanger sequencing; in these cases, it was performed following a positive result of NGS and before testing other family members in order to exclude the possibility of false negatives with the Sanger technique (a phenomenon known as "allelic dropout", caused by the presence of single nucleotide variants in the binding sites of the primers necessary for the PCR step of Sanger sequencing, possibly resulting in the missed amplification of the allele) [9].

## 3. Results

From December 2021 to December 2023, a total of 170 patients were recruited for genetic testing: 163 patients were directly recruited from the cardiologists during their clinical practice, whereas 7 patients had a first contact with the medical geneticist. The indication for NGS made by cardiologists never required additional geneticist visits before performing the genetic testing. The majority of patients ($n = 131$) underwent an NGS analysis, whereas a minority ($n = 29$) only underwent Sanger sequencing. Patients had a mean age of $64 \pm 15$ years and 111 were males (60.3%) (Table 3). A positive family history for cardiomyopathies and/or cardiovascular disorders was present in 81 patients (44%). The clinical suspects prompting NGS analysis were dilated cardiomyopathy (DCM) in 66 patients (38.8%), hypertrophic cardiomyopathy (HCM) in 34 patients (20.0%), arrhythmogenic right ventricular cardiomyopathy (ARVD) in 14 patients (8.2%), cardiac amyloidosis in 26 patients (15.3%), and finally left ventricular non-compaction cardiomyopathy (LVNC) in 3 patients (1.8%). syndrome). Lastly, 27 patients were tested following a relative with a known variant.

As of the 31 December 2023, 125 patients received the result of their genetic investigation, 27 of which consisted of Sanger sequencing of a known variant and therefore had a shorter turnaround time. The mean time for the NGS analysis result was $103 \pm 41$ days from the blood draw to the test result. The mean time from the blood draw to the joint consultation with the medical geneticists and the cardiologist was, instead, $137 \pm 87$ days. Moreover, the elapsed time between submitting the online genetic test request and receiving instructions on the next steps to be taken (either NGS or preliminary genetic counseling) was always less than 3 days.

The test result was positive (i.e., showing at least one C4 or C5 variant) in 21 patients (16.8%). In 48 patients (38.4%), no variants were found. A minority of patients presented with more than one variant: seventeen patients presented with two variants (13.6%); four patients presented with three variants (3.2%), and one patient presented with four (0.8%). Therefore, a total of 78 gene variants were identified; 6 (7.7%) were interpretable as C5, 15 (19.2%) as C4, and 57 (73.1%) as C3-VUS.

As expected based on the literature, the most frequently mutated gene in DCM was *TTN* (three variants). Regarding HCM, the most frequently mutated genes were *MYH7* (three variants) and *MYBPC3* (two variants). One patient with suspected cardiac amyloidosis had a variant in the *TTR* gene. The specific variants classified as C4 and C5 are summarized in Table 4, while Table 5 summarize the positivity rates for our commonest phenotypes comparing C4 and C5 variants alone and plus C3-VUS variants.

**Table 3.** Baseline characteristics of the patients. Abbreviations: DCM, dilated cardiomyopathy; HCM, hypertrophic cardiomyopathy; ARVD, arrhythmogenic right ventricular cardiomyopathy; LVNC, left ventricular non-compaction cardiomyopathy.

|  | Patients (*n* = 170) |
|---|---|
| Age, mean (SD), y | 64 (15) |
| Sex |  |
| Males | 111 (60.3%) |
| Females | 59 (39.7%) |
| Positive family history | 81 (44%) |
| Relative with a known variant | 27 (15.9%) |
| Clinical suspicion |  |
| DCM | 81 (33.8%) |
| HCM | 66 (38.8%) |
| ARVD | 34 (20%) |
| LVNC | 3 (1.8%) |
| Amyloidosis | 26 (15.1%) |

**Table 4.** C4 and C5 variants. Abbreviations: DCM, dilated cardiomyopathy; HCM, hypertrophic cardiomyopathy; ARVD, arrhythmogenic right ventricular cardiomyopathy; LVNC, left ventricular non-compaction cardiomyopathy.

| Clinical Suspicion | Reference Sequence | Mutated Gene | Variant | Interpretation |
|---|---|---|---|---|
| DCM | NM_004281.4 | *BAG3* | c.1534delC p.(Ser513fs*53) | C4 |
|  | NM_005572.3 | *LMNA* | c.569G>A (p.Arg190Gln) | C4 |
|  | NM_000256.3 | *MYBPC3* | c.1224-80G>A | C4 |
|  | NM_000257.4 | *MYH7* | c.2134C>T p.(Arg712Cys) | C4 |
|  | NM_005633.4 | *SOS1* | c.755T>C; p. (lle252Thr) | C4 |
|  | NM_001267550.2 | *TTN* | c.49870C>T; p.(Arg16624*) | C4 |
|  |  |  | c.74724_74730dupTCCTGGT; p.Pro24911fs*23 | C4 |
|  |  |  | c.93166C>T; p.Arg31056* | C4 |
|  |  |  | c.93202G>T; p.(Glu31086*) | C4 |
| HCM | NM_000256.3 | *MYBPC3* | c.1828G>C p.Asp610His | C4 |
|  |  |  | c.3617delG p.Gly1206fs*31 | C4 |
|  | NM_000257.4 | *MYH7* | c.1484T>C; p.(Val495Ala) | C4 |
|  |  |  | c.2123G>C p.(Gly708Ala) | C5 |
|  |  |  | c.2631G>T; p.(Met877Ile) | C5 |
|  | NM_000363.5 | *TNNI3* | c.557G>A (p.Arg186Gln) | C4 |
| ARVD | NM_001458.5 | *FLNC* | c.3937C>T p.(Arg1313*) | C5 |
|  | NM_004415.4 | *DSP* | c.3337C>T [p.(Arg1113*)] | C5 |
|  |  |  | c.3889C>T p.(Gln1297*) | C4 |
|  | NM_004949.5 | *DSC2* | c.268G>T (p.Glu90*) | C5 |
|  | NM_004281.4 | *BAG3* | c.1534delC p.(Ser513fs*53) | C4 |
| Amyloidosis | NM_000371.4 | *TTR* | c.250T>C p.(Phe84Leu) (alias p.Phe64Leu) | C5 |

**Table 5.** Positivity rates. Abbreviations: DCM, dilated cardiomyopathy; HCM, hypertrophic cardiomyopathy; ARVD, arrhythmogenic right ventricular cardiomyopathy.

| Clinical Suspect | Positivity Rate for C4 and C5 Variants | Positivity Rate for C3-VUS, C4 and C5 Variants |
|:---:|:---:|:---:|
| DCM | 26.3% | 63.1% |
| HCM | 26% | 52.1% |
| ARVD | 28.6% | 92.8% |

Patients with C3-VUS will undergo periodic and thorough re-evaluations of the literature along with their clinical cases, with the purpose of better understanding the outcomes of the variants and potentially reclassify them [10].

During the joint consultation with the cardiologist and medical geneticist, the possibility of screening family members with Sanger sequencing was offered to all patients for whom it was possible to reasonably suppose a causative link between the identified variant and the phenotype. For what concerns C3-VUS, since many of them are very unlikely to be relevant to the clinical phenotype, they were not screened with Sanger sequencing in the family members. The multidisciplinary consultation was in these cases crucial to determine whether or not it was appropriate to test the family members by evaluating the phenotype and the family history. To this date, only two families were offered screening for a C3-VUS due to the presence of more than one family member with a phenotype that could be related to the variant previously identified in the patient. Among all patients who underwent Sanger sequencing following the identification of gene variants in affected family members, the majority was identified through the process of familial screening of patients who were already included in this study. This has resulted in 10 families currently being followed up in the context of this cardiogenetic clinic.

## 4. Discussion

The results show how this innovative management model already made it possible to identify an overall considerable number of genetically based cardiomyopathies (78 C3-C4-C5 variants in 170 patients over the time span of approximately 24 months).

In particular, the former procedure for genetic testing referral had several substantial issues. First of all, numerous patients did not comply with the initial referral to the medical geneticist, leading to significant patient dispersion to begin. Second, for those who did go through the process, it inevitably implied a considerable loss of time between the first indication for genetic testing and the results. Moreover, multiple consultations for each patient represented a high burden for the genetic service, and the lack of systematic organization for the whole process made it more difficult to keep track of results and effectively perform family screening.

In contrast, the new way of accessing genetic counseling allows not only a significant decrease in time (with the mean time to obtain the result of NGS just barely exceeding 3 months) but also a remarkable reduction in patient dispersion. This reduction is achieved both in the pre-test phase, as patients receive the indications to perform the blood draw for genetic testing directly by their cardiologists (without the need for other preliminary visits), and in the after-test phase, since the consultation with the medical geneticist is arranged directly with the cardiologists. All in all, this streamlined process allows an effective management of genetic testing and its implications, with a clear optimization of both time and resources.

Furthermore, the standardization of the procedure allows a more systematic recruitment of patients, as opposed to the arbitrary selection which was performed before. All the cardiologists working in the cardiomyopathy clinic are now aware of the precise steps to be taken in case of a suspected genetic cardiomyopathy, ensuring that no patient misses the opportunity to access genetic testing.

Ensuring easy and time-effective access to genetic testing is of crucial importance in the management of patients with cardiomyopathies. Optimizing the genetic testing and

counseling process has allowed for improved clinical management of patients, particularly those for whom the test results have changed the course of treatment—for instance, necessitating the implantation of an ICD. Considering the great impact that these pathologies have on patient's health and quality of life, timely diagnosis and treatment are essential, which underlines the importance of the possible diagnostic implications of genetic testing (i.e., allowing final diagnosis to be reached in patients who present with a borderline phenotype). In fact, it can allow the identification of family members who carry the pathogenic variant and are therefore at risk of developing cardiac disease even before the onset of an overt phenotype and its complications (such as heart failure and cardiac arrhythmias), thus permitting prompt intervention and adequate follow-up. This proactive strategy should ultimately help in reducing the burden of disease on affected families. Moreover, rising evidence is suggesting that genetic data might allow gene-targeted and more personalized therapeutic approaches [11].

The possibility to engage families in the process of genetic testing more effectively is another relevant consequence of the standardized approach that has been implemented, guiding the choice of cascade screening with an integrated cooperation by cardiologists and medical geneticists. Moreover, this continuous collaboration between the two medical specialties—which can be seen as a first attempt towards a real "cardiogenetic clinic"—allows more effective monitoring of the family members who have been tested and their results, ensuring a more comprehensive evaluation of the whole families rather than of family members as individual patients.

The positivity rates vary significantly when considering C4 and C5 variants only and adding C3-VUS. Many patients with highly suggestive phenotypes and no acquired etiology for cardiomyopathy were found to have C3-VUS; therefore, when those are brought into the equation, the positive rates rise significantly, reaching numbers in line with previous studies including C3-VUS in the positivity rates [12]. With the increasing use of NGS, the occurrence of C3-VUS findings is increasing exponentially [13,14], posing a significant challenge in the management of patients with cardiomyopathies. This is due to the fact that finding a C3-VUS on one hand raises the opportunity of discovering a possible cause underlying the patient's condition, which may become relevant with further research in the future. On the other hand, it may have no clinical value and hence should not be used to screen the family or change the course of action for the patient. A thorough evaluation by the medical geneticist and cardiologist is what really makes the difference in the outcome for the patient and his or her family. In particular, what is evaluated is once more the family history, and most importantly whether other close family members display the same clinical phenotype or not, and how this correlates with the presence or absence of the variant in said family members. Specifically, in these cases, cascade genetic screening can be indicated following a clinical screening showing the presence of the phenotype as a tool for familial segregation, i.e., to confirm the presence of a variant in two or more family members presenting with the same phenotype, which can argue in favor of the possible implication of the variant in the pathogenesis of the disease. On the contrary (in accordance with what is recommended by the 2023 ESC guidelines) [15,16], it is not advisable to perform indiscriminate genetic screening of first-degree relatives solely relying on the identification of a C3-VUS.

The assessment of patients presenting with C3 variants carried out mainly by the medical geneticist can potentially lead to different outcomes, depending on whether the clinical phenotype and the family history are strongly suggestive for a pathogenic role of the variant. Despite an initial indication for investigating the genetic background of the patient, during the genetic consultation following the test, the medical geneticist can decide not to proceed with further investigations, notwithstanding the presence of a gene variant (de facto considering the test result as negative). This decision is reached based on the absence of a relevant family history and when the observed C3 variants can be reasonably considered at low risk of being responsible for the phenotype of the patient (based on current available knowledge about the specific gene and variant). On the contrary, when

the personal and familiar background lead the geneticist to offer Sanger sequencing of the known variant in the family members, the cascade of genetic testing can have huge implications for the involved family. Testing for a known variant at a lower cost and with a faster technique also provides an advantage in the management of the family members [17]. Furthermore, it is an example of familial segregation (i.e., the process of testing multiple family members with a certain phenotype in order to try and prove that they all carry the same variant), which can eventually be employed to help the scientific community understand the true pathogenicity of C3-VUSs.

As of now, 10 families are being followed in the context of this "cardiogenetic clinic" in order to manage the overall consequences of carrying pathogenic or likely pathogenic gene variants. Furthermore, in spite of the fact that we emphasize the potential beneficial consequences of familial screening in the case of an identification of pathogenic variants, trying to empower our patients to make informed decision in this sense, some of them were not interested in involving any of their close relatives in the genetic analysis, which represents a limit in the possible impact of genetic testing.

The main limit of our approach is related to the intrinsic nature of the information retrieved by genetic testing, i.e., the uncertainty about the pathogenicity of many identified variants, which sometimes represents a barrier to effective interventions for patients and their families. Nevertheless, significant progress has been made in the field of genetics of cardiovascular diseases over the past years, and further efforts should be continued in order to expand knowledge of genetic testing for cardiomyopathies, with the goal of eventually reaching solid conclusions about the meanings of as many gene and variants as possible.

## 5. Conclusions

In this work, we presented the initial results of an innovative approach to the diagnostic workup of cardiomyopathies, which systematically includes genetic testing early in the diagnostic process and routinely offers genetic and/or clinical screening to family members of patients carrying variants, taking charge of whole families. The creation of this "cardiogenetic clinic" has allowed us to provide answers to our patients in a faster and more effective way while also optimizing our resources, as well as preventing the loss of patients during both the diagnostic process itself and follow-up.

**Author Contributions:** Conceptualization, M.B., N.G. and D.F.G.; methodology, M.B. and N.G.; software, S.D.; validation, M.B., D.F.G. and S.D.; formal analysis, M.B., V.G. and M.L.; investigation, M.B., F.M., G.G., A.H.M. and M.G.; resources, A.C.; data curation, M.L., C.B., G.N., G.M.B.d.P., F.M., G.G., A.H.M. and M.G.; writing—original draft preparation, N.G., V.G. and M.B.; writing—review and editing, M.B., F.M., G.G., A.H.M. and M.G.; visualization, M.B. and N.G.; supervision, D.F.G., S.D. and A.C.; project administration, M.L. All authors have read and agreed to the published version of the manuscript.

**Funding:** This research received no external funding.

**Institutional Review Board Statement:** The study was conducted in accordance with the Declaration of Helsinki and approved by the Institutional Ethics Committee of A.O.U. San Luigi Gonzaga, Orbassano and AA.SS.LL. TO3–TO4–TO5.

**Informed Consent Statement:** Informed consent was obtained from all subjects involved in the study.

**Data Availability Statement:** The raw data supporting the conclusions of this article will be made available by the authors on request.

**Acknowledgments:** The authors thank Nurse Anna Iadevaio for organizing, collecting, and managing the blood samples of all the patients included in the study.

**Conflicts of Interest:** The authors declare no conflicts of interest.

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
