# Peer review of "Genetic Testing for Patients with Cardiomyopathies: The INDACO Study—Towards a Cardiogenetic Clinic"

_cardiogenetics, doi:10.3390/cardiogenetics14030010_

Round 1
Reviewer 1 Report
Comments and Suggestions for Authors
Authors reported a new approach based on available data concerning families diagnosed with an ICC. Some points should be clarified:
- Please detail the number of genes analyzed in each group/cohort. It seems that the number of genes analyzed is not the same.
- All variants were classified following ACMG recommendations? If yes, in which year? What about reclassification of variants, especially if VUS?
- Group of "No variants identified" refers to no rare variant in any of genes analyzed? Or no rare variants in any of genes currently associated with the diagnosed disease?
- All patients showed a definite clinical diagnosis before genetic testing? Or any suspected?
Author Response
Dear reviewer,
We wish to express our gratitude for the time and effort you have put into offering feedback on our article. We considered each of the reviewers' comments and made the necessary adjustments. Below, we have answered each comment and described the changes that were made in response.
Reviewer 1
- Please detail the number of genes analyzed in each group/cohort. It seems that the number of genes analyzed is not the same.
In the methods section, we have addressed that the panel was selected amongst the ones listed in Tab. 1 and Tab. 2 based on a specific clinical suspect.
- All variants were classified following ACMG recommendations? If yes, in which year? What about reclassification of variants, especially if VUS?
The technical specification for the variant classification has been included in the methods section.
- Group of "No variants identified" refers to no rare variant in any of genes analyzed? Or no rare variants in any of genes currently associated with the diagnosed disease?
We use the term "no variants identified" to refer to patients with no variants in the genes that comprise the selected panel. The panels are selected based on the clinical suspect after a thorough clinical evaluation.
- All patients showed a definite clinical diagnosis before genetic testing? Or any suspected?
All patients underwent a thorough clinical workup, which included the application of any clinical examinations eligible to determine a suspected diagnosis. The genetic panel was selected based on the phenotype and clinical suspect.
Reviewer 2 Report
Comments and Suggestions for Authors
Dear Editor,
Bianco et al. provided an interesting manuscript entitled “Genetic testing for patients with cardiomyopathies: the 2 INDACO study - towards a cardiogenetics clinic” in which the authors described their center’s implementation of genetic testing for cardiomyopathies as recommended by current ESC guidelines.
I find manuscript very intriguing and with great medical value, however I have some comments/suggestions that should be addressed before acceptance:
1. The authors conclude that “an innovative approach to the diagnostic workup of cardiomyopathies”. What was the innovation that the authors are writing about? What is so different about their approach to genetic testing than that of other centers? It seems to me that the authors successfully implemented the standard cardiogenetics clinic model.
2. Citation 7 is wrongfully presented and table 3 has an empty cell that maybe should contain DCM? Please also write the abbreviation’s meaning in the table. Please label figure 1 correctly. Is table 2 is completely missing?
3. What was the clinical features of the patients? Please insert a table describing your population. There should be at least age, sex, comorbidities, echo data, familial history, symptoms (NYHA class, syncope, cardiac arrest), ICD implant, CRT rate, ablation therapy etc. Are the baseline characteristics different between C4-C5 versus non C4-C5?
4. How was the differential diagnosis with Brugada syndrome made? ARVD often overlaps with Brugada (please also cite SCN5A positivity also in Brugada).
5. Did the genetic testing result change the clinical approach? Was an ICD implanted or ablation therapy performed after a positive genetic result? Some mutations associate with ventricular arrhythmias which require ablation and ICD (e.g. PKP2 and DSG2 gene mutation associates with ventricular arrhythmias and biventricular ARVD involvement https://doi.org/10.1016/j.amjcard.2022.07.011 ).
Author Response
Dear reviewer,
We wish to express our gratitude for the time and effort you have put into offering feedback on our article. We considered each of the reviewers' comments and made the necessary adjustments. Below, we have answered each comment and described the changes that were made in response.
- The authors conclude that “an innovative approach to the diagnostic workup of cardiomyopathies”. What was the innovation that the authors are writing about? What is so different about their approach to genetic testing than that of other centers? It seems to me that the authors successfully implemented the standard cardiogenetics clinic model.
The primary innovation between our approach and the guidelines' recommendations is the absence of a consultation with the geneticist prior to performing a genetic test, which allowed us to reduce patient management times. The advances presented have been covered in full in the methods section. Moreover, in the Piedmont Region's healthcare reality, a gap has emerged in the availability of specialized cardiogenetics services. The genetics of cardiomyopathies, contrary to other medical sub-specialties, has never had a distinct diagnostic and therapeutic pathway, resulting in it being assimilated into generic practices. However, this model of management can be applied to different conditions, emphasizing the significance of interdisciplinary participation in specialized clinics.
- Citation 7 is wrongfully presented and table 3 has an empty cell that maybe should contain DCM? Please also write the abbreviation’s meaning in the table. Please label figure 1 correctly. Is table 2 is completely missing?
We have completed the necessary formatting changes as suggested, including adding Table 2 to the updated text, fixing Citation 7, accurately labeling the tables, and adding the abbreviations’ meanings.
- What was the clinical features of the patients? Please insert a table describing your population. There should be at least age, sex, comorbidities, echo data, familial history, symptoms (NYHA class, syncope, cardiac arrest), ICD implant, CRT rate, ablation therapy etc. Are the baseline characteristics different between C4-C5 versus non C4-C5?
The clinical description of the population goes beyond the objective of the paper, which is our center's approach to diagnosing genetically based cardiomyopathies. Nevertheless, we added a table describing the baseline characteristics of our patients, which might be of fair interest. We thank the reviewer for the insight regarding the different clinical characteristics of C4-C5 vs non-C4-C5, which could be explored in a following multi-centric study.
- How was the differential diagnosis with Brugada syndrome made? ARVD often overlaps with Brugada (please also cite SCN5A positivity also in Brugada).
The full workup for arrhythmogenic cardiomyopathy includes an MRI and, if necessary, an Ajmaline test, in order to rule out the possibility of Brugada syndrome. A specification on this matter has been included in the methods section.
- Did the genetic testing result change the clinical approach? Was an ICD implanted or ablation therapy performed after a positive genetic result? Some mutations associate with ventricular arrhythmias which require ablation and ICD (e.g. PKP2 and DSG2 gene mutation associates with ventricular arrhythmias and biventricular ARVD involvement https://doi.org/10.1016/j.amjcard.2022.07.011 ).
Some patients' genetic results have influenced how they are clinically managed. A discussion on the clinical approach goes beyond the scope of this work. We thank the reviewer for the insight on the therapeutic impact of genetic testing that gave us the chance to add a specification to the discussion section of this paper.
Please find the revised manuscript.
Should you require any further information or clarification, please do not hesitate to contact us.
Round 2
Reviewer 1 Report
Comments and Suggestions for Authors
No comments
Reviewer 2 Report
Comments and Suggestions for Authors
manuscript is now in acceptable form.